# Infinite-Fidelity Coregionalization for Physical Simulation

**Shibo Li, Zheng Wang, Robert M. Kirby, and Shandian Zhe**
School of Computing, University of Utah
Salt Lake City, UT 84112
{shibo, wzhut, kirby, zhe}@cs.utah.edu

## Abstract

Multi-fidelity modeling and learning is important in physical simulation related applications. It can leverage both low-fidelity and high-fidelity examples for training so as to reduce the cost of data generation yet still achieving good performance. While existing approaches only model finite, discrete fidelities, in practice, the feasible fidelity choice is often infinite, which can correspond to a continuous mesh spacing or finite element length. In this paper, we propose Infinite Fidelity Coregionalization (IFC). Given the data, our method can extract and exploit rich information within infinite, continuous fidelities to bolster the prediction accuracy. Our model can interpolate and/or extrapolate the predictions to novel fidelities that are not covered by the training data. Specifically, we introduce a low-dimensional latent output as a continuous function of the fidelity and input, and multiple it with a basis matrix to predict high-dimensional solution outputs. We model the latent output as a neural Ordinary Differential Equation (ODE) to capture the complex relationships within and integrate information throughout the continuous fidelities. We then use Gaussian processes or another ODE to estimate the fidelity-varying bases. For efficient inference, we reorganize the bases as a tensor, and use a tensor-Gaussian variational posterior approximation to develop a scalable inference algorithm for massive outputs. We show the advantage of our method in several benchmark tasks in computational physics.

## 1 Introduction

Many scientific and engineering applications demand physical simulations, for which the task is mainly to solve partial differential equations (PDEs) at a domain of interest. For example, to estimate the temperature change at the end of a part, one might need to solve transient heat transfer equations over the part (Incropera et al., 2007). Due to the high cost of running numerical solvers, in practice it is often important to train a surrogate model (Kennedy and O'Hagan, 2000; Conti and O'Hagan, 2010). Given the PDE parameters and/or parameterized boundary/initial conditions, we use the surrogate model to predict the high-dimensional solution field, rather than run the numerical solvers from scratch. In this way, we can greatly reduce the cost, because computing the prediction for a machine learning model is often much more efficient and faster.

However, we still have to run the numerical solvers to generate the training data for the surrogate model, which is costly and can be a bottleneck. To alleviate this issue, a wise strategy is to conduct multi-fidelity learning. High-fidelity examples are generated via dense meshes (or smaller finite elements), hence are accurate yet expensive to compute; low-fidelity examples are generated with coarse meshes, which are much cheaper for computation yet quite inaccurate. Despite the (significant) difference in quality, the low-fidelity and high-fidelity examples are strongly correlated since they are based on the same equation(s) or physical laws. Many multi-fidelity surrogate modeling and learning methods have therefore been developed to effectively combine examples of different fidelities to

36th Conference on Neural Information Processing Systems (NeurIPS 2022).

improve the prediction accuracy while reducing the cost of data generation, *e.g.*, (Perdikaris et al., 2017; Parussini et al., 2017; Xing et al., 2021a; Wang et al., 2021).

While successful, the exiting methods only model finite, discrete fidelities, which usually corresponds to several pre-specified meshes (or finite elements). However, since the spacing of the mesh (or the length of finite elements) is continuous, its choice can be infinite and therefore corresponds to infinitely many fidelities. To extract and take advantage of rich information within these infinite, continuous fidelities, we propose IFC, an infinite-fidelity coregionalization method. Our model can flexibly estimate the complex relationships among these fidelities to bolster the predictive performance, and scale to high-dimensional outputs, which are common in physical simulation. Specifically, we first introduce a low-dimensional latent output, which is a continuous function of the input and fidelity. We model the latent output as an ordinary differential equation (ODE), where the dynamics (fidelity derivative) is a neural network with the input as the latent output itself plus the original input, *i.e.*, neural ODE (Chen et al., 2018). In this way, we can capture the complex relationships within and integrate the information throughout the continuous fidelities. To predict high-dimensional outputs, we multiply the latent output with a basis matrix. We place a Gaussian process prior over the basis elements or use another element-wise ODE to capture the basis variations along with the fidelity. For scalable inference of the GP bases, we re-organize the basis matrix as a tensor and introduce a tensor-Gaussian distribution as the variational posterior. In this way, not only can we capture the strong posterior dependency among the massive basis elements, we also avoid estimating the full posterior covariance matrix (which can be huge) and greatly reduce the parameters. We then use the Kronecker product properties and ODE solvers to develop an efficient variational inference algorithm.

For evaluation, we tested our method for predicting the solution fields of three benchmark PDEs, including Poisson's, Heat and Burger's equations. We also applied IFC in topology structure optimization and computational fluid dynamics (CFDs). The output dimension for these tasks varies from thousands to hundreds of thousands. In all the cases, IFC outperforms the state-of-the-art multi-fidelity learning methods by a large margin. In addition, we examined the performance of IFC in making predictions with novel fidelities (other than the training fidelities). It shows that our model with the ODE bases can extrapolate the prediction to new fidelities higher than (*i.e.*, more accurate than) the training fidelities. This opens up a possibility to achieve high-fidelity predictive performance by only using low fidelity data.

## 2 Background

**Linear Model of Coregionalization.** Many tasks demand learning a function of high-dimensional outputs, where the dimension of the input is relatively low. For example, given the scalar viscosity (input), we want to predict the solution of the viscous Burger's equation at a $128 \times 128$ grid on some domain of interest (output). A popular and classical high-dimensional output regression method is Linear Model of Coregionalization (LMC) (Journel and Huijbregts, 1978), which introduces a low dimensional latent output $\mathbf{h}(x) = [h_1(\mathbf{x}), \dots, h_K(\mathbf{x})]^\top$ where each $h_k : \mathbb{R}^s \to \mathbb{R}$ and $s$ is the input dimension. LMC models the actual high-dimensional output $\mathbf{f} \in \mathbb{R}^d$ by linearly combining the latent output elements with a basis matrix $\mathbf{B} = [\mathbf{b}_1, \dots, \mathbf{b}_K]$,

$$\mathbf{f}(\mathbf{x}) = \sum_{k=1}^{K} h_k(\mathbf{x})\mathbf{b}_k = \mathbf{B} \cdot \mathbf{h}(\mathbf{x}) \tag{1}$$

where $K \ll d$ and each $\mathbf{b}_k \in \mathbb{R}^d$. To flexibly estimate each latent output, we often use a Gaussian process (GP) prior (Rasmussen and Williams, 2006). GP is a popular approach to estimate single-output functions. In general, suppose given the training data $\mathcal{D} = \{(\mathbf{x}_1, y_1), \dots, (\mathbf{x}_N, y_N)\}$, we want to learn a function $g : \mathbb{R}^s \to \mathbb{R}$. With the GP prior over $g$, the function values $\mathbf{g} = [g(\mathbf{x}_1), \dots, g(\mathbf{x}_N)]^\top$ follow a multivariate Gaussian distribution, $p(\mathbf{g}|\mathbf{X}) = \mathcal{N}(\mathbf{g}|\mathbf{m}, \mathbf{K})$, where $\mathbf{m}$ is the mean function evaluated at the training inputs, usually set to $\mathbf{0}$, $\mathbf{K}$ is the covariance matrix, and each element $[\mathbf{K}]_{ij} = \kappa(\mathbf{x}_i, \mathbf{x}_j)$ is a covariance (kernel) function of the inputs. The observations $\mathbf{y} = [y_1, \dots, y_N]^\top$ are often assumed to be generated from a Gaussian noise model, $p(\mathbf{y}|\mathbf{g}) = \mathcal{N}(\mathbf{y}|\mathbf{g}, \sigma^2\mathbf{I})$ where $\sigma^2$ is the noise variance. We can marginalize out $\mathbf{g}$ to obtain the marginal likelihood, $p(\mathbf{y}|\mathbf{X}) = \mathcal{N}(\mathbf{y}|\mathbf{0}, \mathbf{K} + \sigma^2\mathbf{I})$. The kernel parameters and noise variance can be estimated by maximizing the marginal likelihood. Due to the Gaussian form, given the new input $\mathbf{x}^*$, the predictive distribution of $g(\mathbf{x}^*)$ is straightforward to compute, which is a conditional Gaussian.

While we can jointly estimate the latent outputs and bases in (1), an effective approach is to conduct Principled Component Analysis (PCA) on the training data to identify the bases $\mathbf{B}$, and then use the singular values as the training outputs to learn the latent functions $h_k(\mathbf{x})$ with standard GP regression. We refer to this method as PCA-GP (Higdon et al., 2008).

**Multi-fidelity Coregionalization.** Practical applications often allow us to collect data with varying fidelities to enable a trade-off between the cost and efficiency. For example, in physical simulation, one can adjust the mesh spacing or length of the finite elements in the numerical solver to generate solution examples at different fidelities. Many multi-fidelity models have been developed to synergize training examples of different fidelities. For example, (Xing et al., 2021a) recently proposed deep residual coregionalization, which sequentially learns $M$ PCA-GP models $l_1, \ldots, l_M$, for the given $M$ fidelities. At each fidelity $m$, it first uses the lower fidelity models to make predictions and then compute the residual error between the low-fidelity predictions and the training outputs at the current fidelity. Based on the residual, it performs PCA to find the bases and estimates the latent output via GP regression,

$$l_m = \text{PCA-GP}(\mathbf{X}_m^{\text{train}}, \mathbf{R}_m^{\text{train}}), \quad \mathbf{R}_m^{\text{train}} = \mathbf{Y}_m^{\text{train}} - \sum\nolimits_{j=1}^{m-1} l_j(\mathbf{X}_m^{\text{train}}), \tag{2}$$

where $(\mathbf{X}_m^{\text{train}}, \mathbf{Y}_m^{\text{train}})$ is the training inputs and outputs at fidelity $m$, and $l_j(\cdot)$ the prediction made by the PCA-GP at fidelity $j$. The prediction at the highest fidelity $M$ is obtained by summing the predictions of all the $M$ models. Other than the sequential training, the recent works of Wang et al. (2021); Li et al. (2022) jointly learn the bases and latent output for every fidelity. To estimate the relationship of successive fidelities, they model the latent output at fidelity $m$ as a nonlinear function of the latent output at fidelity $m - 1$,

$$\mathbf{h}_m(\mathbf{x}) = \boldsymbol{\alpha}\left(\mathbf{h}_{m-1}(\mathbf{x}), \mathbf{x}\right), \quad \mathbf{f}_m(\mathbf{x}) = \mathbf{B}_m \mathbf{h}_m(\mathbf{x}), \tag{3}$$

where $\mathbf{B}_m$ is the basis matrix at fidelity $m$, $\mathbf{h}_m(\cdot)$ and $\mathbf{h}_{m-1}(\cdot)$ are the latent outputs at fidelity $m$ and $m - 1$, respectively, and $\mathbf{f}_m(\mathbf{x})$ is the prediction at fidelity $m$. To fulfill this auto-regression, Wang et al. (2021) proposed a matrix GP prior over $\boldsymbol{\alpha}(\cdot)$ that introduces an additional dependency on the bases, while Li et al. (2022) used a (deep) neural network to model $\boldsymbol{\alpha}(\cdot)$.

## 3    Model

Despite their success, the existing multi-fidelity approaches only model or estimate the relationships between finite, discrete fidelities. In physical simulation, these fidelities often correspond to several specific mesh spacings or finite element lengths. However, since the mesh spacing or element length is continuous, we actually have infinitely many possible choices, which correspond to infinitely many fidelities. Among the continuous, infinite fidelities are much richer information or relationships that can be valuable to promote the predictive performance. To extract and take advantage of this information, we propose IFC, an infinite-fidelity coregionalization model.

Specifically, since the fidelity $m$ can be viewed as continuous (corresponding to the continuous mesh spacing and finite element length), we model the latent output as a continuous function of the input and fidelity, *i.e.*, $\mathbf{h}(\mathbf{x}, m)$. Inspired by the residual coregionalization of Xing et al. (2021a) (see (2)), we model the latent output — which can be viewed as a low-rank summary of the actual high-dimensional output — as the latent output at the preceding (lower) fidelity, plus an adjustment/correction for the current fidelity,

$$\mathbf{h}(m, \mathbf{x}) = \mathbf{h}(m - \Delta, \mathbf{x}) + \boldsymbol{\psi}, \tag{4}$$

where $\Delta > 0$ is an infinitesimal and $\boldsymbol{\psi}$ is the correction term. To capture the complex yet strong relationship with the proceeding fidelity $m - \Delta$, we model $\boldsymbol{\psi}$ as a function of the latent output at $m - \Delta$, the current fidelity $m$, and the input: $\boldsymbol{\psi} = \boldsymbol{\psi}\left(m, \mathbf{h}(m - \Delta, \mathbf{x}), \mathbf{x}\right)$. Since $\lim\limits_{\Delta \to 0} \boldsymbol{\psi} = \mathbf{0}$, it is natural to assume $\boldsymbol{\psi} = \Delta \cdot \boldsymbol{\phi}$. Therefore, we have

$$\mathbf{h}(m, \mathbf{x}) = \mathbf{h}(m - \Delta, \mathbf{x}) + \Delta \cdot \boldsymbol{\phi}\left(m, \mathbf{h}(m - \Delta, \mathbf{x}), \mathbf{x}\right).$$

Moving $\mathbf{h}(m - \Delta, \mathbf{x})$ to the left, dividing the equation by $\Delta$, and taking the limit of $\Delta$ to zero, we arrive at an ODE model,

$$\frac{\partial \mathbf{h}(m, \mathbf{x})}{\partial m} = \boldsymbol{\phi}\left(m, \mathbf{h}(m, \mathbf{x}), \mathbf{x}\right). \tag{5}$$

Without loss of generality, we assume the lowest fidelity is 0. We then model the initial state of the ODE, *i.e.*, the latent output at the lowest fidelity, as a function of the input $\mathbf{x}$,

$$\mathbf{h}(0, \mathbf{x}) = \boldsymbol{\beta}(\mathbf{x}). \tag{6}$$

To flexibly estimate $\boldsymbol{\beta}$ and $\boldsymbol{\phi}$, we parameterize them as neural networks. The advantage of our modeling is two-fold. First, according to (5) and (6), the prediction at an arbitrary fidelity $m$ is $\mathbf{h}(m, \mathbf{x}) = \mathbf{h}(0, \mathbf{x}) + \int_0^m \boldsymbol{\phi}(v, \mathbf{h}(v, \mathbf{x}), \mathbf{x}) \, \mathrm{d}v$, which integrates the predictions from all possible lower fidelities. Thereby, it enables us to exploit information from infinite, continuous fidelities. Second, learning dynamics $\boldsymbol{\phi}$ via neural networks enables us to capture the complex relationships among these continuous fidelities so as to bolster the predictive performance. The above component is an instance of neural ODE (Chen et al., 2018), and a continuous extension of the auto-regressive model in (3).

Similar to LMC (see (1)), we multiply the latent output $\mathbf{h}(\mathbf{x}, m)$ with a basis matrix $\mathbf{B}$ to obtain the high-dimensional output at fidelity $m$. However, the bases can vary along with the fidelity. To flexibly capture such variations, we propose two methods.

**IFC-GPODE** We model each basis element $b_{ij}$ as a function of the fidelity $m$ and place a GP prior,

$$b_{ij}(m) \sim \mathcal{GP}(0, \kappa(m, m')), \tag{7}$$

where $\kappa(\cdot, \cdot)$ is the kernel function. Suppose we have collected a set of training examples $\mathcal{D} = \{(\mathbf{x}_n, m_n, \mathbf{y}_n)\}_{n=1}^N$. We use a Gaussian noise model to sample the observed data. The joint distribution is given by

$$p(\mathcal{B}, \mathcal{Y}|\mathbf{X}) = \prod_{i=1}^d \prod_{k=1}^K \mathcal{N}(\mathbf{b}_{ij}|\mathbf{0}, \mathbf{K}) \prod_{n=1}^N \mathcal{N}(\mathbf{y}_n|\mathbf{B}_n\mathbf{h}(m_n, \mathbf{x}_n), \sigma^2\mathbf{I}) \tag{8}$$

where $\mathbf{X} = \{\mathbf{x}_1, \ldots, \mathbf{x}_N\}$, $\mathcal{Y} = \{\mathbf{y}_1, \ldots, \mathbf{y}_N\}$, $\mathcal{B} = \{\mathbf{b}_{ij}\}_{1 \leq i \leq d, 1 \leq j \leq K}$, $\mathbf{b}_{ij} = [b_{ij}(s_1), \ldots, b_{ij}(s_T)]^\top$ is the basis values at different fidelities, $\{s_j\}_{j=1}^T$ are the distinct fidelities in the data, $\mathbf{K}$ is the kernel matrix on $\{\mathbf{s}_j\}$, and $\mathbf{B}_n = [b_{ij}(m_n)]_{1 \leq i \leq d, 1 \leq j \leq K}$ is the basis matrix at fidelity $m_n$. Note that the latent output $\mathbf{h}(m_n, \mathbf{x}_n)$ is the state of the ODE system in (5) and (6).

**IFC-ODE[2]** Our second method is to model each element $b_{ij}$ with another ODE system,

$$\frac{\partial b_{ij}(m)}{\partial m} = \gamma(b_{ij}, m), \quad b_{ij}(0) = \nu_{ij}, \tag{9}$$

where $\gamma$ is parameterized by a neural network. In this way, we can flexibly capture the evolution of the bases along with the fidelity. The joint distribution is

$$p(\mathcal{Y}|\mathbf{X}) = \prod_{n=1}^N \mathcal{N}(\mathbf{y}_n|\mathbf{B}_n\mathbf{h}(m_n, \mathbf{x}_n), \sigma^2\mathbf{I}) \tag{10}$$

where both $\mathbf{B}_n$ and $\mathbf{h}$ are computed from ODE solvers.

## 4 Algorithm

We now present the inference algorithm. Both IFC-GPODE and IFC-ODE[2] demand we compute the gradient of the learning objective w.r.t to the ODE parameters and initial states, *i.e.*, the parameters for $\boldsymbol{\phi}$ and $\boldsymbol{\beta}$ in (5) and (6) and for $\gamma$ and $\nu_{ij}$ in (9). This can be efficiently done by applying automatic differentiation during the numerical integration in ODE solvers (*e.g.*, the Runge-Kutta method (Dormand and Prince, 1980)). However, the computational graph can be memory intensive. When the memory is insufficient, we can use the adjoint state approach instead (Pontryagin, 1987; Chen et al., 2018), which constructs an adjoint backward ODE system. The gradient is computed by solving the adjoint ODE. We refer to the details in (Chen et al., 2018).

We estimate the parameters of IFC-ODE[2] by maximizing the log joint probability (10) via stochastic optimization, which is relatively straightforward. The learning of IFC-GPODE, however, is much more challenging in that we need to estimate the posterior distribution of the bases at the observed fidelities, $\mathcal{B} = \{\mathbf{b}_{ij}\}_{1 \leq i \leq d, 1 \leq k \leq K}$, which consists of $dKT$ elements. The posterior distribution does not have a closed form, and we resort to the variational inference framework (Wainwright and

Jordan, 2008). Since these bases are coupled in both the GP prior (across the fidelities) and the likelihood (across the outputs), they are strongly dependent in the posterior. Hence, it is natural to introduce a multi-variate Gaussian distribution for $\mathcal{B}$ as the variational posterior to capture these dependencies. However, since the output dimension $d$ is often large, say, hundreds of thousands, the computation and storage of the posterior covariance matrix ($dKT \times dKT$) is prohibitively costly or even infeasible. To sidestep this issue, one might consider the commonly used mean-field variational approximation (Wainwright and Jordan, 2008), which uses a fully factorized posterior. However, doing this will lose all the posterior dependencies and can result in inferior inference quality.

To address this issue, we use an idea of (Zhe et al., 2019; Li et al., 2021) to fold the output space into an $R$ dimensional tensor space, $d_1 \times \ldots \times d_R$ where $d = \prod_{r=1}^{R} d_r$. For convenience, we set $d_1 = \ldots = d_R = \sqrt[R]{d}$. Then we can arrange $\mathcal{B}$ as a $d_1 \times \ldots \times d_R \times K \times T$ tensor. To fully capture the posterior correlations while still achieving a compact parameterization, we introduce a tensor-Gaussian distribution as the variational posterior for the bases $\mathcal{B}$. The tensor-Gaussian is a straightforward extension of the matrix Gaussian distribution,

$$q(\mathcal{B}) = \mathcal{TN}\left(\mathcal{B}|\mathcal{U}, \mathbf{\Sigma}_1, \ldots, \mathbf{\Sigma}_R, \mathbf{\Sigma}_{R+1}, \mathbf{\Sigma}_{R+2}\right) = \mathcal{N}\left(\text{vec}(\mathcal{B})|\text{vec}(\mathcal{U}), \mathbf{\Sigma}_1 \otimes \ldots \otimes \mathbf{\Sigma}_{R+2}\right), \quad (11)$$

where $\mathcal{U}$ is the posterior mean, and $\mathbf{\Sigma}_r$ is the posterior covariance at each mode $r$ ($1 \leq r \leq R+2$). To ensure the positive definiteness, we parameterize each covariance matrix by $\mathbf{\Sigma}_r = \mathbf{L}_r \mathbf{L}_r^\top$ where $\mathbf{L}_r$ is a lower-triangular matrix (*i.e.*, the Cholesky decomposition form). In this way, the total number of parameters for the posterior covariance is reduced to $\sum_{r=1}^{R} \frac{d_r(d_r+1)}{2} + \frac{K(K+1)}{2} + \frac{T(T+1)}{2}$. Consider $d = 10^6$, $K = 10$, and $T = 100$ as an example. If we fold the output into a three-dimension tensor, *i.e.*, $R = 3$, we only need $2 \times 10^{-5} dKT$ parameters to represent the whole $dKT \times dKT$ covariance matrix. Thus, the number of variational parameters is dramatically reduced ($> 99.99\%$) while our variational posterior can still represent the strong posterior dependencies.

We then construct the variational evidence lower bound (ELBO) with the tensor-Gaussian posterior (11), $\mathcal{L} = \mathbb{E}_{q(\mathcal{B})}\left[\log \frac{p(\mathcal{B}, \mathcal{Y}|\mathbf{X})}{q(\mathcal{B})}\right]$. We maximize the ELBO to obtain the variational parameters $\mathcal{U}$ and $\{\mathbf{L}_r\}$, ODE parameters, and noise variance $\sigma^2$. We leverage the Kronecker product properties (Stegle et al., 2011) to decompose the full covariance matrix and to simplify the ELBO,

$$\mathcal{L} = -\text{KL}(q(\mathcal{B})\|p(\mathcal{B})) + \sum_{n=1}^{N} \mathbb{E}_{q(\mathcal{B})}\left[\log p(\mathbf{Y}_n|\mathbf{x}_n, \mathcal{B})\right] \quad (12)$$

where

$$\text{KL}(q(\mathcal{B})\|p(\mathcal{B})) = \frac{1}{2}\text{tr}\left(\mathbf{K}^{-1}\mathbf{\Sigma}_{R+2}\right)\prod_{r=1}^{R+1}\text{tr}(\mathbf{\Sigma}_r) + \frac{1}{2}\text{tr}(\mathbf{K}^{-1}\mathbf{U}_{R+2}\mathbf{U}_{R+2}^\top)$$
$$+ \frac{1}{2}dK\log\det(\mathbf{K}) - \frac{1}{2}\sum_{r=1}^{R+2}\frac{dKT}{d_r}\log\det(\mathbf{\Sigma}_r), \quad (13)$$

$$\mathbb{E}_q\left[\log p(\mathbf{Y}_n|\mathbf{x}_n, \mathcal{B})\right] = -\frac{d}{2}\log(2\pi\sigma^2) - \frac{1}{2\sigma^2}\left(\mathbf{y}_n^\top \mathbf{y}_n - 2\mathbf{y}_n^\top \mathbb{E}_q[\mathbf{B}_n]\mathbf{z}_n + \text{tr}(\mathbb{E}_q\left[\mathbf{B}_n^\top \mathbf{B}_n\right]\mathbf{z}_n\mathbf{z}_n^\top)\right)$$

where $\mathbf{z}_n \triangleq \mathbf{h}(m_n, \mathbf{x}_n)$, $\mathbf{U}_{R+2}$ is obtained by unfolding the mean tensor $\mathcal{U}$ at mode $R + 2$, giving a $T \times dK$ matrix, $\mathbb{E}_q[\mathbf{B}_n]$ is obtained by fetching the $m_n$-th slice of $\mathcal{U}$ at mode $R + 2$ and reshape it as a $d \times K$ matrix, and $\mathbb{E}_q[\mathbf{B}_n^\top \mathbf{B}_n] = \left(\prod_{r=1}^{R}\text{tr}(\mathbf{\Sigma}_r)\right)\mathbf{\Sigma}_{R+1} + \mathbb{E}_q\left[\mathbf{B}_n\right]\mathbb{E}_q\left[\mathbf{B}_n\right]^\top$. The computation is restricted to the covariance matrices at each mode and hence is much more efficient. Note that we can always choose enough large $R$ to ensure $d_R$ is small (*e.g.*, $\leq 100$) so that the computation of the per-mode covariance matrix is efficient and cheap. We can use any gradient-based optimization method to maximize the ELBO.

## 5    Related Work

Linear model of coregionalization (LMC) (Matheron, 1982; Goulard and Voltz, 1992) is a classical framework to extend the standard GP regression for multi-output function estimation. There have been many instances and variants, such as intrinsic coregionalization (Goovaerts et al., 1997), PCA-GP (Higdon et al., 2008), KPCA-GP (Xing et al., 2016), and IsoMap-GP (Xing et al., 2015). GP regression networks (GPRNs) (Wilson et al., 2012) place a GP prior over the basis elements in

LMC and model the bases as functions of the input as well. While more flexible, it brings in additional computational challenges. In addition to LMC, other multi-output regression approaches include convolution GPs (Higdon, 2002; Boyle and Frean, 2005; Alvarez et al., 2019) and multi-task GPs (Bonilla et al., 2007, 2008; Rakitsch et al., 2013). They use kernel convolution and matrix GP priors to model the multiple function outputs. The sparse GP approximations were applied for large output dimensions (Alvarez and Lawrence, 2009; Álvarez et al., 2010). A great survey is given in (Alvarez et al., 2012). The recent work of Zhe et al. (2019) tensorized the output space and learned a set of coordinate features to scale up to massive outputs and to capture the output correlations. To scale up GPRNs to high-dimensional outputs, Li et al. (2021) tensorized the bases and latent output, and developed a structural variational inference with matrix Gaussian and tensor Gaussian posteriors. They also used the Kronecker product properties to simplify the computation. Hence, our approximation technique is similar to these works.

To fulfill multi-fidelity training, Perdikaris et al. (2017); Cutajar et al. (2019) learned a sequence of GP regressors, where each GP is for one fidelity, and models the output as a function of the input and the prediction at the previous fidelity. Their model is an instance of deep GPs (Damianou and Lawrence, 2013; Hebbal et al., 2019). However, their method might not be amenable to a large number of outputs, since these outputs will serve as a part of the input to the GP at the next fidelity, and henceforth greatly increase the input dimension of that GP model. Wang et al. (2021) addressed this issue by fulfilling an auto-regressive structure over the low-dimensional latent outputs. They used a matrix GP prior to sample the latent output as a function of the latent output at the previous fidelity, the input, and the bases. Li et al. (2022) instead used auto-regressive neural networks to model the latent output, and developed an active learning algorithm to dynamically query at new inputs and fidelities. In addition, recently Hamelijnck et al. (2019) developed a multi-resolution, multi-task (output) regression method based on GPRNs and mixtures of experts (Rasmussen and Ghahramani, 2002), which intends to integrate data collected by sensor networks. The network nodes can have multiple resolutions. Other most recent multi-fidelity models include (Wang and Lin, 2020; Wu et al., 2022; Xing et al., 2021b), *etc*. All these works assume the fidelities are fixed and finite, and model the relationship between these discrete fidelities. Our work is different in that we point out the presence and value of continuous fidelities, especially in physical simulation, and we develop a new method to capture and leverage the rich knowledge/relationships within the continuous, infinite fidelities to further enhance the predictive performance.

## 6 Experiment

### 6.1 Predicting Solution Fields of Partial Differential Equations

We first tested IFC for predicting the solution fields of several benchmark PDEs in computational physics, including Poisson's, Heat and Burger's equations (Olsen-Kettle, 2011). To collect the training data, we run the numerical solvers with several meshes. Denser meshes give examples of higher fidelities. The output vector comprises of the solution values on the grid. For instance, a mesh of size $50 \times 50$ corresponds to an output vector of $2,500$ dimensions. For Poisson's and Heat equations, we generated training examples of four fidelities, using $8 \times 8$, $16 \times 16$, $32 \times 32$ and $64 \times 64$ meshes, respectively. For Burger's equation, we used $16 \times 16$, $24 \times 24$, $32 \times 32$ and $64 \times 64$ meshes to generate four-fidelity training data. For all the PDEs, the number of training examples for each fidelity (from the lowest to highest) is $100$, $50$, $20$, and $5$, respectively. For testing, we generated $128$ examples with the highest fidelity. Both the training and test inputs were uniformly sampled from the domain (but non-overlapping). The input includes the parameters of the PDE, the boundary and/or the intial conditions. The input dimension for Poisson's, Heat and Burger's equations is five, three and one, respectively. Hence, the task is in essence to learn an low-to-high mapping that maps the parameters that index a PDE to the solution field of that PDE. The data generation followed the details as provided in (Wang et al., 2021).

**Competing Methods.** We compared with the following state-of-the-art multi-fidelity high-dimensional output learning methods. (1) DRC (Xing et al., 2021a)(`https://github.com/wayXing/DC`), deep residual coregionalization, which performs LMC on the residual error of the predictions from the lower fidelities. The final prediction is the summation of the LMC prediction across all the fidelities. See Sec. 2. (2) MFHoGP (Wang et al., 2021)(`https://github.com/GregDobby/Multi-Fidelity-High-Order-Gaussian-Processes-for-Physical-Simulation`), which uses a matrix GP prior to construct a nonlinear coregionalization (NC) model, and connects multiple

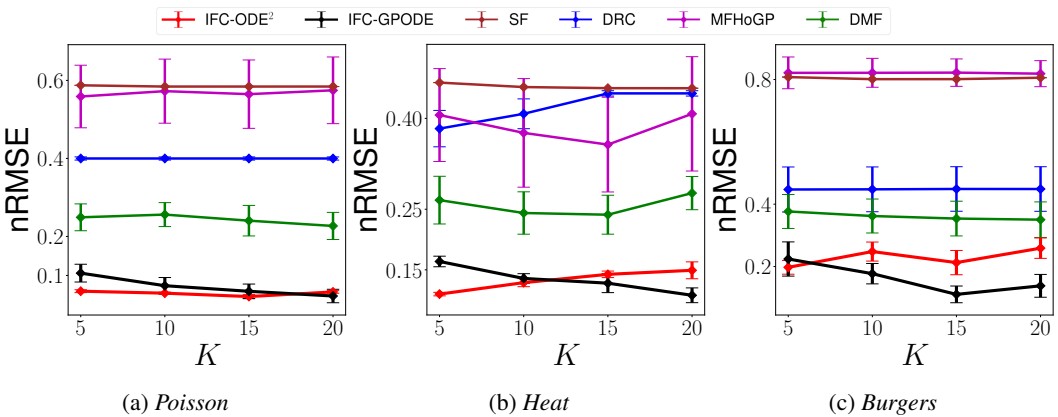

Figure 1: Normalized RMSE in predicting the solution fields of Poisson's, and Heat and Burger's equations. $K$ is the dimension of the latent output.

NC models for multi-fidelity learning, one for each fidelity. To capture the correlation between successive fidelities, the matrix GP prior samples the latent output as a random function of the latent output in the previous fidelity. (3) DMF (Li et al., 2022)(`https://github.com/shib0li/DMFAL`), a neural network (NN) based multi-fidelity learning approach, where each NN models one fidelity. To synergize different fidelities, the latent output of each NN is fed into the NN for the next fidelity. The high-dimensional prediction at each fidelity is obtained through a linear transformation of the latent output. To verify if IFC can indeed better integrate information of distinct fidelities, we also tested (4) SF, the single-fidelity degeneration of our model, where the prediction is $\mathbf{f}(\mathbf{x}) = \mathbf{B}_0 \mathbf{h}_0(\mathbf{x})$, where $\mathbf{B}_0$ is a static basis matrix, and $\mathbf{h}_0(\cdot)$ is a neural network. SF uses all the training examples without differentiation. We denote our ODE based method using the GP prior over the basis matrix by (5) IFC-GPODE and another latent ODE over each basis element by (6) IFC-ODE$^2$.

**Settings and Results.** All the methods were implemented by PyTorch (Paszke et al., 2019), except that DRC was implemented by MATLAB. For our method, we used `torchdiffeq` library (`https://github.com/rtqichen/torchdiffeq`) to solve ODEs and to compute the gradient w.r.t ODE parameters and initial states via automatic differentiation. We used the Runge-Kutta method of order 5 with adaptive steps. For GP related models, including DRC, MFHoGP and IFC-GPODE, we used the square exponential (SE) kernel. The length-scale parameter was initialized to one. For our method, each NN component ($\phi$, $\beta$, and $\gamma$ in Eq. (5), (6) and (9)) employed two hidden layers with `tanh` as the activation function. To handle continuous (infinite) fidelities, we mapped the lowest fidelity to $m = 0$, and highest to $m = 1$. For simplicity, we use a linear mapping from the mesh size to the fidelity value $m$. Suppose the mesh for $m = 0$ is $s_0 \times s_0$, and for $m = 1$ is $s_1 \times s_1$. Then the fidelity of an arbitrary $s \times s$ mesh is $m(s) = (s - s_0)/(s_1 - s_0)$. Note that this is just one way of indexing the mesh size (or spacing) by fidelity values and there can be arbitrary other ways.

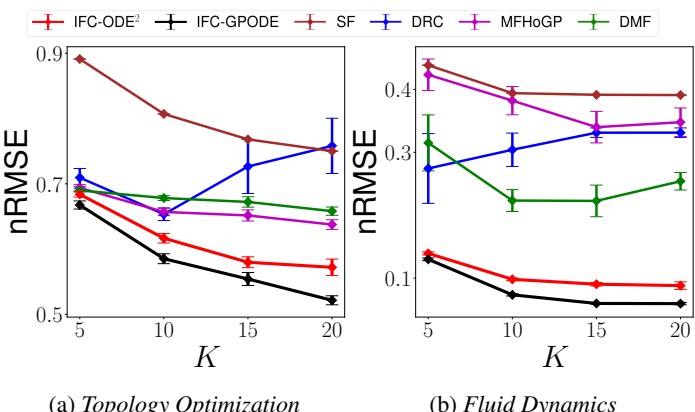

(a) *Topology Optimization*    (b) *Fluid Dynamics*

Figure 2: Normalized RMSE in predicting the optimal topological structures and spatial-temporal pressure field of fluids.

The complex, possibly nonlinear relationships between the fidelities (or meshes) are captured by our neural ODE component (see (5)). Since DRC, MFHoGP and IFC demand the output dimension be the same across different fidelities, we set the output dimension to the one at the highest fidelity (which is $64 \times 64 = 4,096$), and used interpolation

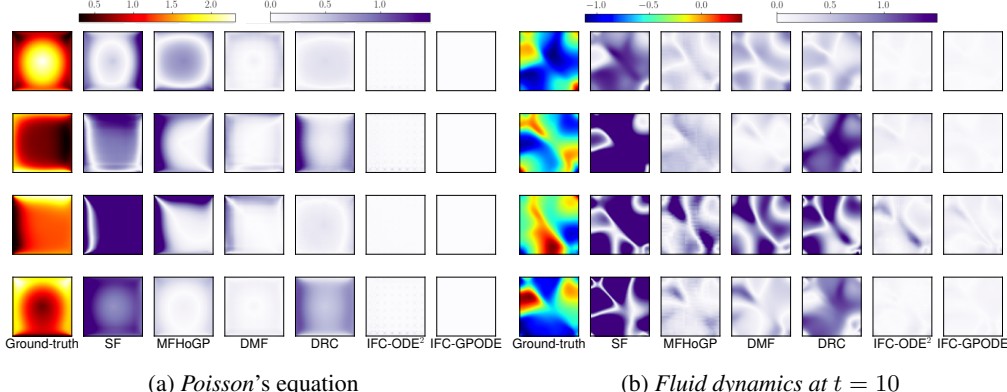

(a) *Poisson*'s equation

(b) *Fluid dynamics at* $t = 10$

Figure 3: Local prediction errors. The leftmost column in (a) and (b) is the original solution. The other columns are the error fields based on the prediction of each method. The lighter the color, the smaller the error.

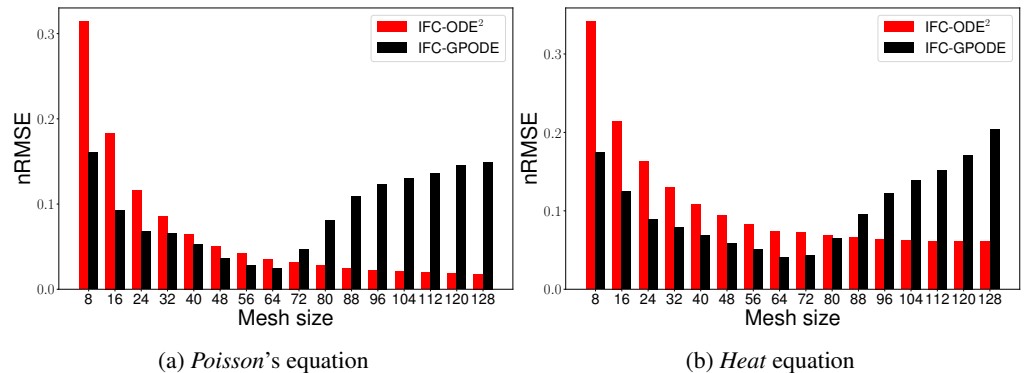

(a) *Poisson*'s equation

(b) *Heat* equation

Figure 4: Normalized RMSE of the predictions with various fidelity values ($m \in [0, 2.14]$). The x-axis shows the corresponding mesh size, where $m = 0$ corresponds to the $8 \times 8$ mesh and $m = 2.14$ the $128 \times 128$ mesh. The largest training fidelity ($m = 1$) corresponds to the $64 \times 64$ mesh.

(or down sampling) to obtain lower dimension predictions to fit the data (Zienkiewicz et al., 1977). For IFC-GPODE, the output is folded into a two-dimensional tensor. For DMF, we also used two hidden layers for each NN, and `tanh` activation, which is consistent with the setting in (Li et al., 2022). The number of neurons per layer was chosen from $\{10, 20, 30, 40, 50, 60\}$. We found that more layers for both our method and DMF did not improve the predictive performance. In addition, other activation functions, such as ReLU and LeakyReLu worsened the performance. This is consistent with the typical choice of the activation function in physics-informed neural networks (Raissi et al., 2019). We ran ADAM (Kingma and Ba, 2014) to train all the models, except DRC that uses L-BFGS to estimate the latent output (the maximum number of iterations was set to 1,000). We used `ReduceLROnPlateau` (Al-Kababji et al., 2022) scheduler to adjust the learning rate from $[10^{-3}, 10^{-2}]$. We set the maximum number of epochs to 5,000, which ensured the convergence of every method. We verified $K$ — the latent output dimension — from $\{5, 10, 15, 20\}$. For each setting, we repeated the experiment for five times. The average normalized root-mean-square-error (nRMSE) and the standard deviation of each method are reported in Fig. 1.

As we can see, both IFC-GPODE and IFC-ODE$^2$ consistently outperform all the competing methods by a large margin. The prediction errors of IFC-GPODE and IFC-ODE$^2$ are much closer, as compared with their difference from the other methods. The both versions of IFC greatly outperforms SF, the single-fidelity degeneration, and in most case SF is also worse than the competing finite fidelity models. This together shows the advantage of our infinite-fidelity modeling, and the improvement is indeed from more effective usage of the training information across dinstinct fidelities.

## 6.2 Topology Optimization

Next, we applied IFC in predicting the optimal topology design structures. Topology optimization (TO) is an important task in engineering design and manufacturing. In general, given the environmental constraint, *e.g.*, an external force, our goal is to find a layout of the give materials (*e.g.*, alloys) that maximizes/minimizes a property of interest, *e.g.*, stiffness. The standard TO solves a constraint optimization problem that includes a compliance objective and total volume constraint (Sigmund, 1997). The computation of the objective often demands for solving associated PDEs, and hence is quite computationally expensive. Hence, we learn a surrogate model to predict the optimal structure outright given the constraint (input). We considered the design problem in (Keshavarzzadeh et al., 2018), which aims to find a structure (discretized in $[0, 1] \times [0, 1]$) with the maximum stiffness under a load on the bottom right half. The load (input) is expressed by the location (in $[0.5, 1]$) and angle (in $[0, \frac{\pi}{2}]$). The strength of the load is fixed. During the optimization, we need to repeatedly run a numerical solver. To learn the surrogate model, we generated training examples with four fidelities, corresponding to $50 \times 50$, $60 \times 60$, $70 \times 70$ and $80 \times 80$ meshes. Again, we generated 100, 50, 20, 5 for each fidelity, and 128 examples at the highest fidelity for testing. We repeated the experiment for five times. The average nRMSE and standard deviation are shown in Fig. 2a. IFC-GPODE and IFC-ODE$^2$ achieve much higher prediction accuracy than all the competing methods in all the cases. It is interesting to see that the performance of our method kept improving with the increase of the latent output dimension. This might be because more latent output elements can summarize and propagate the fidelity information more comprehensively and accurately.

## 6.3 Computational Fluid Dynamics

Third, we applied IFC in predicting the simulation results of computational fluid dynamics. We considered a flow driven by rectangular boundaries (Bozeman and Dalton, 1973). The rectangular is in the domain $[0, 1] \times [0, 1]$. Each of the four boundaries has a prescribed velocity. The spatial-temporal field can be computed by solving the incompressive Navier-Stokes (NS) equations (Chorin, 1968), which is known to be costly due to the complex behaviors under large Renolds numbers. We were interested in predicting the pressure field of the flow along with time in $[0, 10]$, given the Reynolds number in $[10, 500]$. We simulated training examples of four fidelities, with spatial meshes of size $32 \times 32$, $48 \times 48$, $64 \times 64$ and $80 \times 80$ respectively. The number of time steps was set to 20. Hence, the output dimension (at the highest fidelity) is 128,000. Similar to the previous experiments, we collected 100, 50, 20, and 5 examples for each fidelity, and 128 examples at the highest fidelity for testing. We examined the prediction accuracy of each method. For IFC-GPODE, the output is folded as a $20 \times 80 \times 80$ tensor. We repeated the experiment for five times and report the average nRMSE in Fig. 2b. We can see that, consistent with the previous comparison results, IFC (both versions) greatly outperforms all the competing baselines, which confirms the advantage of IFC in predicting complex physical simulation results.

Furthermore, to investigate the local errors in predicting individual solution outputs, we randomly selected four test examples for Poisson's equation and fluid dynamics. We examined the absolute error of each method in predicting every output. For fluid dynamics, we restrict the prediction at $t = 10$. We visualized the error field for each example in Fig. 3 b and c. As we can see, in most cases, the competing methods have dominant errors at several local places, *e.g.*, MFHoGP in Fig. 3a (first three examples) and DRC in Fig.3 b. By contrast, the local errors of IFC-GPODE and IFC-ODE$^2$ are distributed much more uniformly, and much smaller than the competing methods (lighter colors). That means, their performance is much less restricted by a few local regions. This also leads to a better global error.

## 6.4 Interpolation and Extrapolation in Fidelities

Since our method models the output as the function of a continuous fidelity $m$, it can predict the solution outputs at arbitrary $m$ that is different from the training fidelity values, *i.e.*, interpolation and extrapolation. Note that the current finite, discrete fidelity approaches cannot make such predictions. To examine the performance of our model in interpolating and extrapolating the fidelity of prediction, we tested on Poisson's and Heat equations. We generated four-fidelity training data, including 256, 128, 64, and 32 examples for the first, second, third and fourth fidelity, respectively. The corresponding mesh size is $8 \times 8$, $16 \times 16$, $32 \times 32$, and $64 \times 64$. The lowest fidelity is $m = 0$, and highest $m = 1$. We set the latent output dimension to 20 and trained our model accordingly.

We then used the model to predict the solution at a variety of $m$ values, which corresponds to new meshes. For example, $m = 1.29, 1.57, 2.14$ correspond to meshes of size $80 \times 80$, $96 \times 96$ and $128 \times 128$, respectively. We viewed the "gold-standard" solution as solved with the $128 \times 128$ mesh, under which we generated 256 test examples. We varied $m \in [0, 2.14]$, and examined the corresponding prediction errors as compared with the gold-standard solution. The results are reported in Fig. 4. As we can see, within the range of training fidelities, *i.e.*, $0 \leq m \leq 1$ and the corresponding mesh size less than $64 \times 64$, the prediction error of IFC-GPODE is consistently smaller than that of IFC-ODE$^2$, especially at very low fidelities (*e.g.*, the $8 \times 8$ grid). IFC-GPODE achieves the smallest prediction error at $m = 1$, *i.e.*, the highest training fidelity. When $m > 1$ (mesh size bigger than $64 \times 64$), the performance of IFC-GPODE drops. By contrast, while when $m < 1$, the prediction error of IFC-ODE$^2$ is slightly worse than IFC-GPODE, when $m > 1$, the performance of IFC-ODE$^2$ keeps improving; it achieves the smallest error at the largest $m$ (*i.e.*, $m = 2.14$ corresponding to the $128 \times 128$ mesh), which is smaller than the prediction at $m = 1$ (*i.e.*, highest training fidelity). The nRMSE of IFC-ODE$^2$ at $m = 1$ and $m = 2.14$ is 0.036 *vs.* 0.018 and 0.074 *vs.* 0.061, for Poisson's and Heat equations, respectively. The results show that IFC-GPODE is better in interpolation but IFC-ODE$^2$ is promising in extrapolation. This might be attributed to the GP used IFC-GPODE, which is known to interpolate well yet not good at extrapolation (Rasmussen and Williams, 2006). The improved extrapolation performance of IFC-ODE$^2$ can be particularly useful in practice. It allows us to train the surrogate model only using lower fidelity examples, but we can still expect to gain higher fidelity predictions, *i.e.*, more accurate than the training data. Therefore, we can avoid generating very high-fidelity examples for training to further reduce the cost.

One major limitation of IFC is that the training is much slower than the other methods. For example, on the dataset for Poisson's equation, the average per-epoch/-iteration time of DRC, MFHoGP, DMF, IFC-GPODE and IFC-ODE$^2$ is 0.02, 1.05, 0.04, 4.95 and 7.84 seconds, respectively ($K = 20$). For the fluid dynamics, the average per-epoch/-iteration time is 0.04, 1.28, 0.10, 14.65 and 21.54 seconds, respectively ($K = 20$). This mainly arises from the intensive computation in back-propagating the gradient throughout the numerical integration in the ODE solver. One might improve the speed by using lower order solvers or larger step-sizes, which, however, can hurt the accuracy of the gradient computation. Note that, after training, the prediction of IFC is instantly fast (as fast as the competing methods), because simply doing numerical integration is very efficient.

## 7 Conclusion

We have presented IFC, an infinite coregionalization method for physical simulation. Through ODE based modeling, our method can capture and integrate information from infinite, continuous fidelities to facilitate learning. Our algorithm can scale up to high-dimensional outputs. The experimental results have shown an encouraging improvement upon the existing finite, discrete fidelity methods. In the future, we plan to develop an active learning scheme for our model to further reduce the training data amount and to maximize the benefit-cost ratio.

## Acknowledgments

This work has been supported by MURI AFOSR grant FA9550-20-1-0358 and NSF CAREER Award IIS-2046295.

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
